Ontogenetic feeding shifts in two thresher shark species in the Galapagos Marine Reserve

http://orcid.org/0000-0001-7756-7564 Arnés-Urgellés Camila 1 2
Galván-Magaña Felipe 1
Elorriaga-Verplancken Fernando R. 1
http://orcid.org/0000-0002-7240-1570 Delgado-Huertas Antonio 3
http://orcid.org/0000-0002-2446-9888 Páez-Rosas Diego 2 4 5 dpaez@usfq.edu.ec
1 Centro Interdisciplinario de Ciencias Marinas, Instituto Politécnico Nacional , La Paz, Baja California Sur , Mexico
2 Galapagos Science Center, Universidad San Francisco de Quito , Isla San Cristóbal, Islas Galápagos , Ecuador
3 Instituto Andaluz de Ciencias de la Tierra (CSIC-UGR), Universidad de Granada , Granada, Granada , Spain
4 Fundación Conservando Galápagos, Galapagos Conservancy , Isla Santa Cruz, Islas Galápagos , Ecuador
5 Oficina Técnica San Cristóbal, Direccion Parque Nacional Galápagos , Isla San Cristóbal, Islas Galápagos , Ecuador
Doğdu Servet
Electronic publication date: 2024 Dec 16
Publication date: 2024
Volume: 12
Electronic Location ID: e18681
Received 2024 Aug 15; Accepted 2024 Nov 19
Copyright: © 2024 Arnés-Urgellés et al.
Copyright year: 2024
Copyright holder: Arnés-Urgellés et al.
License: This is an open access article distributed under the terms of the Creative Commons Attribution License, which permits unrestricted use, distribution, reproduction and adaptation in any medium and for any purpose provided that it is properly attributed. For attribution, the original author(s), title, publication source (PeerJ) and either DOI or URL of the article must be cited.
License URL: https://creativecommons.org/licenses/by/4.0/

Keywords: Feeding strategies, Isotopic niche, Ontogenetic changes, Tropical Eastern Pacific, Thresher sharks

Funding: USFQ Collaboration Grant and Galapagos Science Center Grant POA 2019 The Galapagos Science Centre (GSC) Centro Interdisciplinario de Ciencias Marinas (CICIMAR) Instituto Andaluz de Ciencias de la Tierra (IACT-Granada) Consejo Nacional de Humanidades, Ciencias y Tecnologías (CONAHCYT) Instituto Politécnico Nacional (COFAA and EDI) This work was supported by the USFQ Collaboration Grant and Galapagos Science Center Grant (POA 2019). The Galapagos Science Centre (GSC), the Centro Interdisciplinario de Ciencias Marinas (CICIMAR) and the Instituto Andaluz de Ciencias de la Tierra (IACT-Granada) provided the necessary facilities and guidance to complete the laboratory phase of this study. Camila Arnés-Urgellés received fellowship from the Consejo Nacional de Humanidades, Ciencias y Tecnologías (CONAHCYT), which made the successful completion of this work possible. Fernando R. Elorriaga-Verplancken and Felipe Galván-Magaña received fellowships from the Instituto Politécnico Nacional (COFAA and EDI). The funders had no role in study design, data collection and analysis, decision to publish, or preparation of the manuscript.

==============================
Background

The morphology and hunting behavior of thresher sharks make them easily distinguishable. These species are distributed across the Tropical Pacific Ocean feeding on squid and small fish. However, ontogenetic changes in their feeding strategies and habitat use are still unknown in this region.

Methods

We examined the δ13C and δ15N signatures in vertebral collagen from populations of Alopias pelagicus and Alopias superciliosus inhabiting the Galapagos Marine Reserve, focusing on three maturity stages: neonate, juvenile and adult. The vertebrae samples were taken from the seizure of illegal fishing activities carried out by a foreign fleet within the Galapagos archipelago. A total of thirty-three vertebrae from A. pelagicus and twenty-one from A. superciliosus were analyzed.

Results

Both species displayed significant differences in their δ15N values (p < 0.001), but not in δ13C (p = 0.230), suggesting a similar habitat use, but different prey consumption. Throughout their ontogeny, A. pelagicus displayed isotopic differences (p < 0.001), where neonates showed lower δ13C values and higher δ15N values compared to juveniles, probably because they still reflect the isotopic signatures of their mothers even after the first year of life. This study highlights trophic differences between both species, accompanied by an ontogenetic variation in A. pelagicus, aspects that allow us to understand the role of these species within the dynamics of the Eastern Tropical Pacific ecosystem.

Introduction

Habitat utilization and movement of marine vertebrates are mainly driven by ontogenetic shifts in their life-history priorities, such as survival, growth, and reproduction (Heupel, Carlson & Simpfendorfer, 2007; Carlisle et al., 2015). This premise, coupled with intraspecific and interspecific interactions, influences marine trophic dynamics, ecosystem structure, and overall biodiversity (Werner & Gilliam, 1984; Baum & Worm, 2009). Thus, ontogenetic shifts have been studied in elasmobranchs because of their predatorial roles in most marine ecosystems (MacNeil, Skomal & Fisk, 2005; Estupiñán-Montaño et al., 2019). In recent years, these studies have emphasized the necessity of implementing science-based conservation management strategies (Shiffman et al., 2021; Cerutti-Pereyra et al., 2022), as large pelagic fish populations have increasingly decreased, along with the negative impacts of this phenomenon across numerous trophic levels (Baum & Myers, 2004; Ferretti et al., 2010; Bird et al., 2018).

Thresher sharks (Alopias spp) inhabit tropical and subtropical waters worldwide (Compagno, 1984). These species have an elongated dorsal lobe on their caudal fin, nearly as long as their body (Compagno, 1984), which is an essential part of their hunting behavior (Sepulveda et al., 2005; Oliver et al., 2013). The overexploitation of thresher sharks has threatened their survival (Worm et al., 2024), consequently, these species have been listed under Appendix II of the Convention on International Trade in Endangered Species of Wild Fauna and Flora (CITES). The International Union for Conservation of Nature (IUCN) red list assessment also classifies thresher shark populations as overexploited, leading to a decline in the global population (Rigby et al., 2019; Worm et al., 2024). Currently, in the Ecuadorian Pacific, thresher sharks represent more than 70% of the total shark catch (Raharjo, Hartati & Redjeki, 2024), even reaching catches exceeding 150,000 individuals over the last years (Briones-Mendoza, Mejía & Carrasco-Puig, 2022).

Despite this overexploitation, there is scarce knowledge about the life history of thresher sharks in the Eastern Tropical Pacific. A low number of studies have examined their feeding behavior in Ecuadorian waters (Polo-Silva et al., 2007; Polo-Silva, Rendón & Galván-Magaña, 2009; Polo-Silva et al., 2013; Páez-Rosas et al., 2018; Calle-Morán & Galván-Magaña, 2020). However, most of the research has focused on pelagic threshers (Alopias pelagicus), due to their higher landing quota, which facilitates for the collection of samples from fishing ports (Martínez-Ortíz et al., 2015; Briones-Mendoza, Mejía & Carrasco-Puig, 2022). Therefore, most of these studies have been conducted in mainland Ecuador, with the exception of Páez-Rosas et al. (2018), who based their research on pelagic threshers caught illegally in the Galapagos Marine Reserve (GMR). The GMR has some of the largest global shark aggregations (Salinas-de-León et al., 2016; Acuña-Marrero et al., 2018), and maintains relatively intact food webs that support the presence of these species (Páez-Rosas et al., 2024). Consequently, this region becomes an exceptional environment for shark research.

Stable isotope analysis (SIA) permits the quantification of changes in trophic interactions during different periods of development based on the isotopic turnover rate of the tissue (Bearhop et al., 2004; Hussey et al., 2012). Tissues with relatively low isotopic turnover rates (e.g., vertebral collagen) allow the inference of ontogenetic changes in foraging patterns in elasmobranchs (Estrada et al., 2006; Carlisle et al., 2015); since collagen is a metabolically inert tissue and is not resorbed after deposition (Campana, Natanson & Myklevoll, 2002). Therefore, successive layers of different density (i.e., annual groups of growth layers) reflect the conditions under which they were secreted (Campana, Natanson & Myklevoll, 2002), allowing to obtain a summary of feeding history and migratory behavior of these predators (Carlisle et al., 2015; Shen et al., 2022). This technique is based on the fact that isotopic signatures in consumer tissues are related to diet and habitat use (Michener & Schell, 1994; Bearhop et al., 2004). The δ13C values were assessed based on the principle that primary productivity, dissolved CO2 concentration, algal diversity, and other physicochemical processes create a pronounced coastal-oceanic gradient, resulting in a decrease in δ13C in offshore habitats (Michener & Schell, 1994; France, 1995; Newsome et al., 2007). And the principle that the δ15N of tissues increase as one ascends the trophic level owing to enrichment in the consumer’s δ15N relative to its prey (Adams & Sterner, 2000; Post, 2002).

Although ontogenetic studies of sharks in the GMR have been conducted (Estupiñán-Montaño et al., 2019; Salinas-de-León et al., 2019; Páez-Rosas et al., 2021; Cerutti-Pereyra et al., 2022), there is still a lack of understanding about the trophic ecology of sharks in this region. Therefore, this study aims to provide isotopic information differentiated by species, sex and maturity stages that allows understanding the feeding patterns of two thresher shark species (A. pelagicus and A. superciliosus) that inhabit the GMR. Furthermore, our work contributes to existing knowledge regarding the ecology of sharks in the Tropical Eastern Pacific and providing unprecedented insights into the trophic ecology of thresher sharks in the GMR.

Materials and Methods

Study area

The Galapagos Islands are in the Eastern Tropical Pacific Ocean, ~1,000 km from mainland Ecuador. This island complex is home to the GMR, which is limited by a strip of 40 nautical miles, measured from a “baseline” that surrounds the archipelago and its internal waters, generating a protected surface of ~138,000 km2 (Heylings, Bensted-Smith & Altamirano, 2002) (Fig. 1). Its remote location renders it susceptible to the impact of several oceanic currents, which, together with other environmental factors, contribute to their high endemism levels (Edgar et al., 2008; Salinas-de-León et al., 2020). The GMR is influenced by four primary oceanic currents from different directions (i.e., Perú, Panamá, Equatorial, and Sub-equatorial or Cromwell currents), which are responsible for important upwellings (Palacios et al., 2006; Schaeffer et al., 2008; Forryan et al., 2021), and the existence of a strong seasonality (Sweet et al., 2007).

Figure 1 Map of the Galapagos Islands with the boundaries of the Galapagos Marine Reserve.

The mark signals the location where the illegal fishing fleet was taken for inspection at San Cristobal Island.

Sample collection

This research was undertaken under permits: PC-86-19 and was carried out following the protocols of ethics and animal handling approved by the Galapagos National Park Directorate and Ecuadorian laws.

On September 3, 2019, the Galapagos National Park officials, in collaboration with the Ecuadorian Navy, intercepted five illegal fishing boats within the limits of the GMR. These vessels were detained and transported to the nearest port, Puerto Baquerizo Moreno, on the San Cristobal Island. After an identification of the captured individuals, it was determined that there were over 300 individuals of Alopias spp. among the five illegal boats, some of which could not be identified down to species level or maturity size due to their condition. For this study, 33 individuals of A. pelagicus and 21 of A. superciliosus were selected based on their preservation state, specifically selecting those that had a complete head and tail fin. Sex was recorded and total length (TL) data were taken to estimate precaudal length (PCL) based on established relationships. Finally, the first dorsal vertebra of each individual was collected. All remaining shark materials were then destroyed, as required by the Ecuadorian laws.

Sample processing

The cleaning process for vertebrae involved the use of solvents to remove the neural arch and connective tissue. Subsequently, the vertebrae were sanded and polished before being placed in paper bags for drying. Each bag was labeled and stored in the Galapagos Science Center of the Universidad San Francisco de Quito. The diameter of each vertebrae was measured to determine its radius, which allowed the establishment of maturity stages (i.e., neonate, juvenile and adult) based on age and growth information developed by Liu et al. (1999) for A. pelagicus and Liu, Chiang & Chen (1998) for A. superciliosus. Subsequently, samples of vertebral collagen were collected from each growth section, using a micro drill and a 0.7 mm drill bit. Three sets of collagen samples were collected from each vertebra, provided that the individual had reach sexual maturity. However, if the individual was a juvenile, only two samples were obtained. In the case of A. pelagicus, 23 samples were gathered from mature adults and 14 from juveniles, whereas for A. superciliosus, 13 samples were obtained from adults and eight from juveniles.

Vertebral collagen samples were transferred to the Stable Isotope Biochemistry Laboratory of the Andalusian Institute of Earth Sciences IACT (CSIC-URG) in Granada, Spain, where they were weighed using an analytical balance to ensure precision. Samples weighing between 0.6 and 1 mg were stored in silver capsules and subjected to a hydrochloric acid steam bath to degrade any inorganic carbonates that could contaminate the analysis. This process was performed in a desiccator for 24 h. After being treated with the acid, the silver capsules were taken to the isotope ratio mass spectrometer (DELTA plus XL; Thermo Finnagen, Waltham, MA, USA) with 0.1‰ error, in order to quantify the δ13C and δ15N values. The results, expressed in parts per thousand (‰), were obtained using the equation: δ13C or δ15N = 1000([Rsample/Rstandard] − 1), in which Rsample and Rstandard are the 13C/12C or 15N/14N ratios of the sample and the standard, respectively. The standards used were Pee Dee Belemnite for δ13C and atmospheric N2 for δ15N. Finally, we measured the weight percentage of carbon and nitrogen concentrations in each sample and used the C/N ratio as a proxy for protein content (Logan et al., 2008).

Data analysis

Shapiro-Wilk, Kolmogorov-Smirnov, Bartlett and Levene’s tests were employed to assess the normality and homogeneity of the data for both species. Then, Welch’s t-test, Student’s t-test, ANOVA, and Tukey’s test were applied to determine differences between species, sexes, and maturity stages. All statistical analyses were performed in R software using a significance level of p = 0.05.

The Bayesian standard ellipse areas (SEA) were used to estimate isotopic niche width and overlap among thresher sharks’ groups using the package SIBER (Stable Isotope Bayesian Ellipses in statistical software R) (Jackson et al., 2011). This Bayesian method provides a measure of the isotopic niche area at the population level, expressed as the SEA in units of area (‰2) and contains 95% of the data for each analysed group. Monte Carlo simulations were employed to correct the bivariate ellipses (δ13C and δ15N) surrounding the data points in the 95% confidence interval for the distributions of both isotopes (Jackson et al., 2011). The magnitude of the isotopic overlap (‰2) among species, sex and life stages were estimated using the estimations of the ellipses via maximum probability methods (Jackson et al., 2011).

Results

The maximum length recorded for A. pelagicus and A. superciliosus were 2.99 and 2.86m respectively, while the C/N ratio ranged from 3.22‰ to 3.56‰ (Table 1), corroborating that the signatures were within the theoretical range established for the assimilation of protein (McConnaughey & McRoy, 1979).

Table 1 δ13C and δ15N signatures (expressed as ‰; mean ± SD, and C/N relation) of A. pelagicus and A. superciliosus in the Galapagos Marine Reserve.

Species	Sex/Life stage	δ13C (mean ± SD)	δ15N (mean ± SD)	C/N	
A. pelagicus		−14.7 ± 1.18‰	9.5 ± 1.18‰	3.44	
	Females	−14.5 ± 0.8‰	9.9 ± 0.7‰	3.49	
	Males	−14.7 ± 0.7‰	9.5 ± 1.2‰	3.34	
	Neonates	−15.0 ± 1.0‰	9.3 ± 1.0‰	3.56	
	Juveniles	−14.8 ± 0.5‰	9.5 ± 1.3‰	3.43	
	Adults	−14.3 ± 0.5‰	9.7 ± 1.1‰	3.22	
A. superciliosus		−14.9 ± 1.6‰	8.0 ± 1.31‰	3.56	
	Females	−14.3 ± 0.4‰	7.6 ± 1.4‰	3.40	
	Males	−14.6 ± 1.0‰	8.1 ± 1.3‰	3.47	
	Neonates	−14.5 ± 1.2‰	8.6 ± 0.8‰	3.53	
	Juveniles	−14.6 ± 0.7‰	7.8 ± 1.4‰	3.49	
	Adults	−14.4 ± 0.7‰	7.7 ± 1.4‰	3.41	
Note:

Isotopic signatures are categorized by sex and life stage in both species.

Isotopic comparison between species, sex and maturity stages

All isotopic values of both species were analyzed (Table 1), and interspecific differences were observed in δ15N signatures (t = 6.4, df = 98.07, p < 0.001) but not in δ13C signatures (t = −1.2, df = 94.58, p = 0.23). No differences were found in the δ15N and δ13C signatures between sexes for both species (p > 0.05). However, intraspecific differences were observed in δ13C signatures between maturity stages of A. pelagicus (F (2.73) = 4.61, p = 0.01), but not in δ15N signatures (F (2.73) = 0.44, p = 0.64). For the maturity stages of A. superciliosus there were no differences in δ13C (F (2.48) = 0.32, p = 0.72) and δ15N (F (2.48) = 2.42, p = 0.10) signatures. Given the dissimilarities in the δ13C signatures of the maturity stages of A. pelagicus, Tukey’s test was performed; showing that there are differences between the neonate vs. adult stages (−0.65‰, p = 0.01), and between juvenile vs. adult stages (−0.5‰, p = 0.04).

Isotopic niche width and overlap

The SIBER test showed an isotopic niche overlap of 46% between both species (Fig. 2), which can be considered moderate, since A. pelagicus showed higher δ15N values than A. superciliosus (Fig. 2). A similar pattern was identified between sexes in both species (Fig. 3), where there was a 44% overlap between females and males of A. pelagicus was observed, while the overlap between females and males of A. superciliosus was 31%. However, the corrected standard ellipse (SEAc) of females of A. pelagicus was wider (5.3‰2) than that of males (3.3‰2) (Fig. 3 and Table 2), demonstrating some trophic flexibility in females. For A. superciliosus it was the opposite, with females showing a narrower isotopic niche (1.4‰2) compared to males (3.9‰2) (Fig. 3 and Table 2).

Figure 2 Isotopic niche area (δ13C and δ15N) of A. pelagicus and A. superciliosus in the Galapagos Marine Reserve.

Figure 3 Isotopic niche area (δ13C and δ15N) of A. pelagicus and A. superciliosus in the Galapagos Marine Reserve.

The ellipses area shows the degree of overlap within sex groups.

Table 2 Total isotopic area (TA‰2), standard ellipse area (SEA‰2) and corrected standard ellipse (SEAc‰2) recorded for A. pelagicus and A. superciliosus in the Galapagos Marine Reserve.

Species	Sex/Life stage	TA (‰2)	SEA (‰2)	SEAc (‰2)	
A. pelagicus					
	Females	9.8	4.7	5.3	
	Males	15.9	3.3	3.4	
	Neonates	32.2	10.1	10.4	
	Juveniles	15.7	3.8	4.0	
	Adults	6.0	1.9	2.0	
A. superciliosus					
	Females	2.3	1.3	1.4	
	Males	15.8	3.8	3.9	
	Neonates	14.8	5.5	5.8	
	Juveniles	10.0	3.2	3.4	
	Adults	7.1	3.2	3.3	
Note:

Values are categorized by sex and life stage in both species.

Bayesian ellipses demonstrate that neonates of A. pelagicus exhibited a 54% overlap with respect to juveniles and a 33% overlap with adults, whereas juveniles and adults shared an overlap of 45% (Fig. 4). In contrast, the neonates of A. superciliosus displayed a 47% overlap with juveniles and a 40% overlap with adults, while juveniles and adults had an 80% overlap (Fig. 4). Finally, corrected standard ellipse (SEAc) of all age categories in both species, showed that the isotopic niche of neonates is broader than that of juvenile and adults (Table 2).

Figure 4 Isotopic niche area (δ13C and δ15N) of A. pelagicus and A. superciliosus in the Galapagos Marine Reserve.

The ellipses area shows the degree of overlap within age categories in both species.

Discussion

Feeding patterns of thresher shark species

The δ13C and δ15N signatures suggests a consistent pattern of habitat use by A. pelagicus in the GMR throughout their life. This species is known to prefer offshore ecosystems, making its δ13C signatures depleted (Polo-Silva et al., 2013). The populations of A. pelagicus inhabiting mainland Ecuador have more impoverished δ13C signatures (~2‰) than those inhabiting the GMR (Polo-Silva et al., 2013; Páez-Rosas et al., 2018), suggesting the use from oceanic areas where there is less supply of nutrients. Although the populations of A. pelagicus that inhabit the GMR could have a more coastal strategy, it must also be considered that the shorelines of archipelago are characterized by having mangrove ecosystems (Moity, Delgado & Salinas-de-León, 2019). In general, mangrove-dominated shorelines are rich in suspended particulate matter, dissolved organic matter and nutrients (Cawley et al., 2012), which could be enriching the δ13C signatures of A. pelagicus in the GMR, and in turn favoring its permanence in this region.

The δ15N signatures of A. pelagicus at the mainland Ecuador and GMR do not present major differences (Polo-Silva et al., 2013), so both populations could be consuming similar prey throughout the equatorial Pacific. The diet of A. pelagicus in mainland Ecuador consists of three main prey: red flying squid, Ommastrephes bartramii; Humboldt squid, Dosidicus gigas; and purpleback flying squid, Sthenoteuthis oualaniensis (Calle-Morán & Galván-Magaña, 2020), species that are widely distributed throughout the Eastern Tropical Pacific (Galván-Magaña et al., 2013; Yu, Chen & Liu, 2021). Typically, A. pelagicus migrates vertically during the day, spending most of the daylight period at 200–300 m in the mesopelagic zone, and most of the night at 50–150 m in the epipelagic zone (Andrzejaczek et al., 2022). This vertical migration pattern suggests that A. pelagicus may actively pursue prey (i.e., during the squid’s dial migrations) (González-Pestana et al., 2019), however, It is possible that also feeds during the day at greater depths, following prey to their deeper locations (Arostegui et al., 2020).

There were no isotopic differences between males and females of A. pelagicus, suggesting that both sexes preferred similar habitats and feeding sources. This finding coincides with previous studies conducted on this species in Ecuadorian Pacific (Polo-Silva et al., 2013; Páez-Rosas et al., 2018), as well on the west coast of the Baja California Peninsula, Mexico (Sánchez-Latorre et al., 2023). However, our results showed a moderate overlap between sexes, where females had broader isotopic niches, so they could be using other habitats unlike males. This condition could be related to the fact that females need to expand their feeding areas to consume a high amount of key nutrients, such as fatty acids and proteins (Liu et al., 1999), that are crucial for reaching sexual maturity and maintaining the energy required for gestation (Liu et al., 1999). This species also exhibits ovofagia, a process that directs its energy to produce fewer but larger and more mature embryos, which decreases the risk of predation (Miller, Wails & Sulikowski, 2022). Therefore, this reproductive strategy requires a lot of energy, which explains why A. pelagicus females consume prey from different habitats, such as deep-sea squids (Polo-Silva et al., 2013).

Prior research has examined the diet of A. superciliosus in the Ecuadorian Pacific (Polo-Silva et al., 2007; Polo-Silva, Rendón & Galván-Magaña, 2009). However, these studies were conducted in mainland waters, making the current research in baseline information for the GMR. Our isotopic signatures suggest the exploitation of inshore habitats with high productivity within the GMR. A characteristic that coincides with the dietary references of A. superciliosus, where they mention the consumption of benthic and coastal fish (Polo-Silva et al., 2007; Polo-Silva, Rendón & Galván-Magaña, 2009). Thus, based on isotopic signatures of both Alopias species we can infer that there is no interspecific competition, despite using the same habitat within the GMR. This condition would be supported by the Bayesian ellipses, which demonstrated a moderate overlap between both species. Therefore, the consumption of squids by A. pelagicus would cause it to migrate frequently to pelagic ecosystems to feed (Polo-Silva et al., 2013; Calle-Morán & Galván-Magaña, 2020), while A. superciliosus would look for more coastal food sources (Polo-Silva et al., 2007; Polo-Silva, Rendón & Galván-Magaña, 2009).

There were also no isotopic differences were found between males and females of A. superciliosus, indicating a similar use of food sources. Polo-Silva et al. (2007) established that female A. superciliosus remain in offshore areas, but can migrate to inshore ecosystems to feed, while males remain in coastal areas due to a more specialized diet. Stomach content analysis has revealed a more diverse diet in females (22 prey) compared to males (13 prey); where both sexes have preference for demersal fish, however, female preferences also extended to deep-water squids (Polo-Silva et al., 2007; Polo-Silva, Rendón & Galván-Magaña, 2009). Therefore, it is likely that A. superciliosus females migrate to offshore ecosystems more frequently than males, particularly during gestation and parturition periods. The isotopic niche had a low overlap between sex, confirming a differential pattern in habitat use and potential prey. This could imply that although females and males coexist in the same environment, they maintain varied feeding patterns as a strategy to prevent potential intraspecific competition (Briones-Mendoza, Carrasco-Puig & Toala-Franco, 2021).

Ontogenetic feeding patterns

Neonates of A. pelagicus exhibited a more extensive isotopic niche breadth than other groups. The observed variation in δ13C may reflect not only the environmental conditions experienced by the neonates but also the maternal signal or a combination of influences. It is generally challenging to assess δ13C and δ15N signatures in the early life stages of sharks, unless the rate of maternal isotopic signal loss is measured for each species (Olin et al., 2011; Broadhurst et al., 2019). However, there is evidence that at 1 year of age they would have already lost their mother’s δ13C and δ15N signatures, reflecting a condition of independent consumer (Olin, Shipley & McMeans, 2018; Páez-Rosas et al., 2021).

Juvenile sharks exhibited a more restricted habitat usage than neonates, which indicates a shift from opportunists’ habits in neonates to more specialized habits as they grow. Juvenile sharks prioritize meeting their high energy demands and invest this energy in growing at a high rate to reach sexual maturity (Bethea, Buckel & Carlson, 2004; Crear et al., 2021). However, in early stages, pelagic sharks do not yet exhibit specific discrimination between prey, so they could consume coastal prey from lower trophic levels (e.g., mollusks and arthropods) and higher ones (e.g., pelagic and benthic fishes) (Bethea, Buckel & Carlson, 2004). Studies on the identification of juvenile sharks in the GMR have revealed that A. pelagicus is typically found near coasts and bays, always as solitary individuals in apparent feeding activities (Llerena-Martillo, Peñaherrera-Palma & Espinoza, 2018).

Adult and juvenile A. pelagicus demonstrated remarkably similar isotopic averages, indicating comparable habitat use and prey consumption patterns. However, when evaluating the isotopic niche areas, juveniles exhibited a larger amplitude compared to adults. This difference is attributed to the consumption of lower trophic level prey by juveniles and their more generalized habits compared to adults (Polo-Silva et al., 2013; Sánchez-Latorre et al., 2023). In adults the reproduction becomes the most important energetic target (Sims, 2005), so physiological adaptations of A. pelagicus also change as they grow, allowing them to exploit deeper ecosystems (Andrzejaczek et al., 2022). This condition allows them to prey at higher trophic levels, such as cephalopods (O. bartramii, D. gigas, and S. oualaniensis), which comprise 90% of the diet of adult A. pelagicus from the Equatorial Pacific (Galván-Magaña et al., 2013; Calle-Morán & Galván-Magaña, 2020).

No isotopic differences were detected among maturity stages (i.e., neonates, juveniles, and adults) in A. superciliosus, however, isotopic niche areas suggest different feeding patterns for each stage. Neonates of A. superciliosus exhibited a wide isotopic niche indicating the utilization of offshore and inshore habitats. However, this isotopic niche breadth may be related to the isotopic contribution of the mother, which suggests that neonates’ isotopic signatures of A. superciliosus are affected by maternal inputs, even after the first year of life (Hussey, MacNeil & Fisk, 2010; Olin et al., 2011).

Juveniles displayed a reduction in isotopic niche breadth associated with a change in their feeding patterns, specifically in inshore environments. In contrast to the neonate stage where the primary goal is to avoid predators, the juvenile stage is characterized by a high growth rate (Branstetter, 1990). Therefore, this greater need for energy leads them to consume high-calorie prey such as the coastal fish L. argentus and M. gayi (Polo-Silva et al., 2007; Polo-Silva, Rendón & Galván-Magaña, 2009). Juveniles of A. superciliosus spend more time foraging in shallow water during the day and in deeper water at night, whereas adults engage in the reverse pattern (Fernandez-Coelho et al., 2015). Therefore, it is possible that the difference in the use of shallow and deep habitats (Fernandez-Coelho et al., 2015) is a strategy for avoiding potential competition between individuals of the same species.

The juvenile and adult stages of A. superciliosus revealed a high degree of overlap, this finding suggests that individuals of different sizes and sexes share the similar habitats. However, as A. superciliosus progresses through different maturity stages, its feeding strategies become increasingly specialized and focused on inshore habitats (Preti et al., 2008). Even research on vertical migration of this species has shown that adults can dive to depths up to 700 m, where they complement their diet with myctophids and squids (Nakano et al., 2003; Coelho, Fernandez-Carvalho & Santos, 2015).

Conclusions

Lamniformes are known to be present in coastal and oceanic ecosystems. However, unlike A. pelagicus and A. vulpinus, A. superciliosus inhabits continental slopes, which distinguishes it from other Alopias species (Estrada et al., 2003; Smith et al., 2008). The outcomes of this research contribute to the knowledge on the trophic ecology of thresher sharks in the GMR and provide data to establish their contribution within the ecological dynamics of this region. However, it is challenging to accurately assess the status of Alopias populations in the Ecuadorian Pacific, and to develop effective management strategies for this species. Therefore, it is crucial to continue generating information to develop sustainable fisheries and effective conservation measures for these populations.

Supplemental Information

Supplemental Information 1 Dataset.

We thank all park rangers and volunteers who contributed to the sample collection, especially Sarah Beauvais for her help in processing the samples. We also thank the reviewers who helped significantly improve this manuscript.

Additional Information and Declarations

Competing Interests

Author Contributions

Animal Ethics

Data Availability

The authors declare that they have no competing interests.

Camila Arnés-Urgellés conceived and designed the experiments, performed the experiments, analyzed the data, prepared figures and/or tables, authored or reviewed drafts of the article, and approved the final draft.

Felipe Galván-Magaña conceived and designed the experiments, authored or reviewed drafts of the article, and approved the final draft.

Fernando R. Elorriaga-Verplancken performed the experiments, authored or reviewed drafts of the article, and approved the final draft.

Antonio Delgado-Huertas performed the experiments, authored or reviewed drafts of the article, and approved the final draft.

Diego Páez-Rosas conceived and designed the experiments, analyzed the data, authored or reviewed drafts of the article, and approved the final draft.

The following information was supplied relating to ethical approvals (i.e., approving body and any reference numbers):

This research was carried out following the protocols of ethics and animal handling approved by the Galapagos National Park and the Ecuadorian laws.

The following information was supplied regarding data availability:

The raw data is available in the Supplemental Files.

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
