# Peer review of "Ontogenetic feeding shifts in two thresher shark species in the Galapagos Marine Reserve"

_PeerJ, doi:10.7717/peerj.18681_

## Round 0.1 · original submission · Major Revisions

Dear Authors,

Your article has been evaluated by several referees and a ‘Major revision’ decision has been made. Please review the referee's suggestions and corrections carefully.

You can see the reviewers' comments on your article below.

·

Basic reporting

The manuscript is interesting because it describes the feeding ontogeny of two species of thresher sharks in the Galapagos Marine Reserve. This area has characteristics and, is different from the surrounding areas of Ecuador. The information is relevant to see what happens and how these species behave in this region. Although it is very similar to what has been reported by other authors who have worked with the two species in other areas of the Pacific, I consider that there are some interpretations and deficiencies that should be addressed and adjusted in a better way.

Experimental design

They need to improve some paragraphs in this section and I proposed to use another one. Please check the specific comments in additional comments

Validity of the findings

The data analyses and the discussion is in general well but some sections must to be improve or adjusted

Additional comments

I suggest the following recommendations that should be incorporated for the manuscript to be approved after a second review.


Specific Comments
• Lines 41-43. Why estimate the trophic position using SIA in vertebrae? This results what does mean? These values are too low compared with other studies and the information that exists about these species even in other regions of Ecuador and in tropical and subtropical areas…. Be careful with this
• Lines 70-72. Even this is the main species of Ecuador in landing volume and exportation. In the last revision of the trade database of CITES the information of the last 5 years of the shark species listed in Appendix II this is the species No 1 represents more than 70%.
• Lines 75-76. Why not include Polo-Silva et al. 2009?
• The introduction is too long. Please summarize
• Line 181. to estimate isotopic niche why use the SEA and not the SEAc this is the value corrected by the script. Please review this because the value shown by the SEA does not consider the size of the sample
• Lines 191- 200, I do not agree that vertebral tissue is adequate to estimate the trophic position of a species, first because it is necessary to have tissues similar to those of prey and the most common in the constitution is muscle and on the other hand it is the muscle that accumulates the greatest amount of protein that comes from food, therefore this is a more precise tissue that reflects the trophic dynamics of a species in terms of its trophic position. On the other hand, when applying this equation where the enrichment value comes from, this value should be specific to the type of tissue and if possible, estimated in chondrichthyans. I think that this part should be adjusted and removed from the manuscript, it may be showing erroneous information by underestimating the PT of a species that we know is a 3rd and possibly 4th order predator in most of the areas where it lives.
• Lines 204-205. Add a reference about the C/N ratio is into the range proposed.
• Line 220. Change the word breadth by width
• Line 226. In this section report the SEAc but this is not was mentioned in the Material and Methods section. Please check this in this section
• Lines 231-235. Why not use the NicheRover Script? it allows you to estimate the overlap between groups in both ways because generally, the % overlap varies
• Lines 236-239. These lines look like part of the discussion. Please check this
• Lin 242. The TP for A. superciliosus with a value of 2.6 does not make sense. Please review my comments 7 above
• Line 254. Please review the acronyms in some parts where you put GMR and other RMG
• Lines 273-277. It is curious, that you report that according to the result of the Trophic Level, it is categorized as a secondary tertiary carnivore, and you cite Calle and Galvan 2020 and Polo et al. 2013. But although they describe that they do feed on prey with these categories and that they are the prey that can have these trophic levels (~2.6) the TN reported for A. pelagicus is very different from the one you report. That is also associated with the results that in these studies were estimated with what was found in the stomach contents and isotopic analysis in muscle... these results are being influenced by the tissue turnover rate and stating that the TN of this species in the GMR is 2.6 is an error. This is something that should be adjusted or eliminated from the manuscript.
• Lines 305-307. Please this must adjust according to what I mentioned in the previous comment and comment 7… Both species have similar TP in much of the areas where they live, independent that they consumed different prey and few of them similar
• Lines 326-327. How do you know that the signal of neonates reflects the use of inshore and offshore areas?? These values could reflect the maternal signal or even a mix.
• Lines 332-333. What do you expect? Those results showed the normal ontogenetic developments such as must be. I suggest removing these lines.
• Figure 5, I suggest removing it. Fig 4 is more than enough.

Reviewer 2 ·

Basic reporting

iii

Experimental design

Line. 123-125. This island complex is home to the GMR, which is limited by a strip of 40 nautical miles, measured from a “baseline” that surrounds the archipelago and its internal waters, generating a protected area of ~138,000 km2 (Heylings et al., 2002) (Figure 1). The surface is measured in km, but the area should be in km2.


Line. 144-148. For this study, 21 individuals of A. superciliosus and 33 individuals of A. pelagicus were selected based on their size and preservation state. Please, to give the details about the scale of the preservation state that was considerate for this purpose.
Vertebral samples were collected from the anterior dorsal parts of the sharks, and the total length (TL), precaudal length (PCL), and sex were recorded. All remaining shark materials were then destroyed, as required by the Ecuadorian laws. Why did the authors decide to use this measurement? The should be clear that this measurement was used to obtain the TL toward an estimation PCL-TL.

Line 153-156. The diameters of the vertebrae were measured to determine the radius, which allowed for the calculation of maturity stages using the equations developed by Liu et al. (1999) for A. pelagicus and Liu et al. (1998) for A. superciliosus. This is impossible because in both studies these authors try about age, growth and maturity, but in any moment they speak about the pahse of the life cycle.


Line 178-179. Bartlett and Levene’s tests were employed to assess the homogeneity of variances in the different data groups. First, the normality of the data should be done. Then, homogeneity of variances

Line 180-183. To determine statistical differences between species, sexes and life stages, Welch's t-test, Student's t-test, ANOVA, and Tukey’s test for multiple comparisons were applied to identify differences between the subgroups of each species. It is very important to know that this parametric tests should be carried out depending on data have normality and homogeneity of variances.

Validity of the findings

Line 242-244 These results could suggest a possible use of the same ecosystems and prey; however, the higher δ15N signatures of the neonates could be due to another isotopic sources. This part should be in the Discussion section.

Line. 249-151. The trophic position of A. pelagicus is TP = 3.3, placing it as a tertiary consumer within the GMR, while for A. superciliosus it is TP = 2.6 placing it as a secondary consumer within the GMR. These values are very low for these two shark species. Generally, these species occupy trophic levels from four to five in marine and oceanic ecosystems.

Line. 262-263. Although the populations of A. pelagicus that inhabit the RMG could have a more coastal strategy. Please, put GMR, not RMG

Line 271-275. The diet of A. pelagicus in mainland Ecuador consists of 4 main prey: red flying squid, Ommastrephes bartramii;, giant squid Dosidicus gigas, purpleback flying squid Sthenoteuthis oualaniensis, and South Pacific hake Merluccius gayi (Calle-Morán & Galván-Magaña, 2020). Please, the authors review the rules in the scientific writing. Put a comma after the common name and, point and comma then of the scientific name to separate all the species.

Line. 287- 288 There were no isotopic differences between males and females of A. pelagicus, suggesting that both sexes preferred similar habitats and sources. Source of what??? Of feeding
Line 295- 298. This is a pattern observed in the west coast of Baja California Peninsula, Mexico (Sánchez-Latorre et al., 2023). Condition that would be since females need to expand their feeding areas to consume a high amount of nutrients that allow them to reach maturity and maintain the energy required for gestation (Liu et al., 1999). Please cite what will be the most important nutrients to get the sexual maturity and then the gestation.

Line. 353-354. However, in early stages, pelagic sharks do not yet exhibit specific discrimination between prey as they consume items from various trophic levels (Bethea et al., 2011). Indicate which trophic levels the use in their feeding, e.g., trophic levels 3 and 4.

Line. 357. aggregating behavior in this region (Llerena et al., 2013). Please cite very well, the correct one is Llerena-Martillo et al., 2018.

Line.396-398 Adult individuals were classified as secondary consumers, unlike what was reported by Polo-Silva et al., (2013), who, through the analysis of stomach contents and stable isotopes in muscle tissue, placed them as tertiary consumers. Please, correct this beacause the sharks are tertiary or quaternary consumers. Review the research: Calle-Morán and Galván-Magaña (2020) and Calle-Morán et al. (2023). Therefore, based on the number of the samples,it is less wise to confirm it.

Additional comments

Line. 410-412. Therefore, it is crucial to continue generating information to develop sustainable fisheries and effective conservation measures for these populations. The discussion are very poor due to the uthors only try about Ecuador, but what happen in other part of the World. For example: Estrada et al. (2003) with the Alopias vulpinus, etc. Open the discussion to other Lamniformes shark species regarding to stable isotopes. Try to imporve this section with more literature.

Reviewer 3 ·

Basic reporting

Only minor suggested edits for grammar and language and literature references. All other criteria are met:

There are some areas of the manuscript where the sentence structure and grammar need to be improved.

Line 112: Replace the period with a comma. Should be one sentence and read as " ... in the GMR, Including ..."

Line2 212-214: Should read as "For the age categories of A. superciliosus, there were no significant differences in δ13C (F(2, 48) = 0.32, p = 0.72) or δ15N signatures (F(2, 48) = 2.42, p = 0.10)."

Line 248: should be "their life" or "it's life", not "his life"

Line 254: GMR misspelled as RMG

Line 264: D. gigas english name is more commonly referred as "Humbolt squid". Calling it "giant squid" can confuse readers with the much larger species Architeuthis dux.

Line 286 - 289: This sentence reads very awkwardly, consider rewriting.

Line 390: Needs a period after "species".

Experimental design

No comment

Validity of the findings

The most important edits are in regards to the references in the discussion. Some of the statements the authors are stating are not supported by the references they cite:

Line 298 - 301: This seems wrong. This sentence is stating that benthic fish was the main prey item in A. superciliosus according to the studies they cite. González-Pestana et al. 2019 reported squid D. gigas as the most important prey item in both A. superciliosus and A. pelagicus. I don't see any mention of A. superciliosus diet in Polo-Silva et al. 2013.

Lines 309 - 310: I do not find this statement to be supported by the reference given.

Lines 358-359: I did not find mention of "observed ontogenetic differences in the vertebrae of A. superciliosus" in Polo-Silva et al. 2013. I can only find results from A. pelagicus in that paper.

Lines 379 - 380: Again I don't see this information on A superciliosus in Polo-Silva et al. 2013. It would be more appropriate to cite the study that recorded A. superciliosus at those depths. Was it from a telemetry study or was it from catch data? Throughout the whole discussion it seems that this sentence and many others such cite Polo-Silva et al. 2009 "Descripción de la dieta de los tiburones zorro (Alopias pelagicus) y (Alopias superciliosus) durante la época lluviosa en aguas
ecuatorianas."

Additional comments

This paper provides solid insight on the trophic positions and feeding ecology of two thresher shark species from the GMR. I find their analyses to be sound and their conclusions appropriate, with minor exceptions listed below.

I feel that the title of this paper should be changed to include mention of the trophic positions. The major findings from this study were 1) A. pelagicus undergo a ontogenetic shift (which is reflected in the title) and 2) A. pelagicus and A. superciliosus occupy different trophic levels (not reflected in the title). I suggest the title "Trophic positions and ontogenetic feeding shift in two thresher shark species in the Galapagos Marine Reserve"

Other comments:

Lines 271-273: I would caution using the phrase "nocturnal feeding". We know that A. pelagicus conduct diel vertical migration (as is mentioned in the text), likely to follow prey such as squids during their respective diel vertical migration. If they follow the prey to the deeper depths during the day, it is possible that they feed on them down there as they do in the shallower depths at night. In Hawaii, A. pelagicus make brief, frequent excursions towards warmer, shallower depths (50-100m), likely to rewarm their bodies for greater hunting at the colder 200-300m depths where temperatures are around 15 degrees.

Reviewer 4 ·

Basic reporting

In the study, the authors investigated the ontogenetic feeding shifts of two thresher shark species (Alopias pelagicus and A. superciliosus) in the Galapagos Marine Reserve. I present my general criticisms of the major errors that I have identified for the authors to consider below:

First of all, the fact that the vertebrae were obtained from illegal fishing activities indicates that the sample may not have been selected randomly and may not be sufficient to reflect the natural dynamics of the population. This situation seriously limits the generalizability of the results of the study. Individuals obtained from illegal fishing may create a selection bias since they include certain age groups or individuals that are more easily caught during the sampling process.

In addition, only 33 Alopias pelagicus and 21 Alopias superciliosus vertebrae were examined, and this sample size is insufficient to statistically demonstrate ontogenetic changes and trophic position differences. Especially considering the separation of life stages into three stages, sufficient data may not have been obtained from each life stage. Geographical and temporal factors were not taken into account, and details were not provided on where and when the samples were collected; this situation may prevent correct interpretation of habitat use and feeding strategies.

Although the results of isotope analyses are presented, it is not clearly addressed how these analyses are affected by seasonal and geographical variations. Finally, the methodological soundness of the study can be questioned because of the lack of a control group and the fact that samples from non-illegal fishing were not compared. These shortcomings weaken the ecological contribution of the study and raise serious doubts about the reliability of its results.

Altogether, I think the article is not suitable for publication.

Experimental design

The fact that the samples were obtained from illegal fishing activities means that the sample selection was not random. This causes serious limitations in generalizing the results of the study. Illegal individuals may not be representative of the natural population and may create selection bias.

The 33 and 21 vertebrae samples analyzed are not a large enough sample size to understand the ontogenetic and trophic differences among shark species. I think that the number of samples taken from each life stage is insufficient in terms of statistical power, especially considering that the life stages are divided into three.

Since the samples obtained from illegal fishing were used in the study, it is necessary to compare the samples with a reliable control group. I believe that not making a comparison with samples obtained from legal fishing or other methods weakens the accuracy and reliability of the results.

The exact coordinate ranges from which the samples were obtained are not specified. Sampling at different locations may show significant changes in habitat use and trophic dynamics. Analyses performed without controlling for these factors reduce the reliability of the results.

Isotope analysis can provide information about the feeding habits of sharks, but it does not seem to have been considered that isotope signals may vary depending on seasonal changes or geographical factors. In addition, more detailed interpretations are needed about how long isotopic signals from mother sharks are visible in the offspring.

Validity of the findings

As previously mentioned, the article's experimental design, data collection methods, and the generalizability of the analyses diminish the robustness of the findings and cast doubt on its reliability. Therefore, I regret to say that it should be rejected.

---

## Round 0.2 · accepted · Accept

Dear Authors,

Thank you for your submission to PeerJ.
Congratulations, your article has been accepted for publication.

All the best
Servet

·

Basic reporting

The authors incorporated all my suggestions and welcomed the recommendations to remove sessions that did not allow the manuscript to be published. Additionally, I was able to confirm that the recommendations of at least two other reviewers were accepted and therefore I consider that the manuscript can be accepted for publication.

Experimental design

Adjusted

Validity of the findings

Adjusted and logic

Reviewer 3 ·

Basic reporting

No comment

Experimental design

No comment

Validity of the findings

No comment

Additional comments

Line 265: Would be better read as "...making the current research form baseline information ..."

Lines 339 - 341: Is this statement supported by Preti et al. 2008?

Reviewer 4 ·

Basic reporting

All my opinions and suggestions are in section 4.

Experimental design

All my opinions and suggestions are in section 4.

Validity of the findings

All my opinions and suggestions are in section 4.

Additional comments

I expect the authors to understand my extremely important criticisms and concerns stemming from the number of samples and illegal sampling I directed to them while reviewing the article. I find it useful to remind you that these reviews are not for polemic purposes. I would like to thank the authors for accepting my criticisms and for the kind explanations they provided to each of them. I also observed that they revised their articles in line with both my and the other reviewers' evaluations.
Therefore, I think this article is acceptable for publication in its final version.